# Fecal Microbiota Transplantation as New Therapeutic Avenue for Human Diseases

**DOI:** 10.3390/jcm11144119

**Published:** 2022-07-15

**Authors:** Manuele Biazzo, Gabriele Deidda

**Affiliations:** 1The BioArte Limited, Life Sciences Park, Triq San Giljan, SGN 3000 San Gwann, Malta; m.biazzo@thebioarte.com; 2SienabioACTIVE, University of Siena, Via Aldo Moro 1, 53100 Siena, Italy; 3Department of Biomedical Sciences, University of Padua, Via U. Bassi 58/B, 35131 Padova, Italy

**Keywords:** fecal microbiota transplantation, gut microbiome, neurological disorders

## Abstract

The human body is home to a variety of micro-organisms. Most of these microbial communities reside in the gut and are referred to as gut microbiota. Over the last decades, compelling evidence showed that a number of human pathologies are associated with microbiota dysbiosis, thereby suggesting that the reinstatement of physiological microflora balance and composition might ameliorate the clinical symptoms. Among possible microbiota-targeted interventions, pre/pro-biotics supplementations were shown to provide effective results, but the main limitation remains in the limited microbial species available as probiotics. Differently, fecal microbiota transplantation involves the transplantation of a solution of fecal matter from a donor into the intestinal tract of a recipient in order to directly change the recipient’s gut microbial composition aiming to confer a health benefit. Firstly used in the 4th century in traditional Chinese medicine, nowadays, it has been exploited so far to treat recurrent *Clostridioides difficile* infections, but accumulating data coming from a number of clinical trials clearly indicate that fecal microbiota transplantation may also carry the therapeutic potential for a number of other conditions ranging from gastrointestinal to liver diseases, from cancer to inflammatory, infectious, autoimmune diseases and brain disorders, obesity, and metabolic syndrome. In this review, we will summarize the commonly used preparation and delivery methods, comprehensively review the evidence obtained in clinical trials in different human conditions and discuss the variability in the results and the pivotal importance of donor selection. The final aim is to stimulate discussion and open new therapeutic perspectives among experts in the use of fecal microbiota transplantation not only in *Clostridioides difficile* infection but as one of the first strategies to be used to ameliorate a number of human conditions.

## 1. Introduction

### 1.1. The Gut Microbiota

The human body is home to a variety of microbial species (ranging from archaea to fungi and viruses) that form complex microbial communities called *microbiota* interacting with each other and also with our body. The microbial community of each area of the human body is unique in its composition, it presents different microbial ecological niches [1], and it plays an essential role in the general physiological functions and health of an individual [2,3] and is implicated in human diseases [4,5]. Over 98% of the human microbiota is located within the gastrointestinal tract (GI) and is referred to as *gut microbiota*. These micro-organisms collectively represent a dynamic population of microbes (approximately 10^14^ cells) forming a symbiotic superorganism containing 100 times the number of genes of the human genome and weighing approximately the same as the human brain.

The general composition of the gut microbial community included the five phyla *Bacteroidetes*, *Firmicutes*, *Actinobacteria*, *Proteobacteria*, *Verrucomicrobia* [6] with the anaerobic *Bacteroidetes* and *Firmicutes* contributing to more than 90% of the total bacterial species in the healthy gut. The ratio between the two main phyla can change from one individual to another because of (i) differences in individual (host) genomes, and (ii) environmental factors (antibiotic use, lifestyle, hygiene, and diet) [6]. Dysbiosis of the human gut microbiome is associated with a wide range of pathologies, including obesity [7], diabetes [8], diarrhea [9], and irritable bowel syndrome [10]. The concept of microbial dysbiosis also includes the microbiome bacteriophage components that are also implicated in a wide range of physiological (health) and pathological conditions [11].

Importantly, the gut microbiota is not an isolated community that simply lives in the gut and within the host, but it does profoundly communicate with the other organs (even distant ones) using microbial signals transmitted across the intestinal epithelium and via different pathways, including (i) the trimethylamine (TMA)/trimethylamine N-oxide (TMAO), (ii) the short-chain fatty acids (SCFAs), (iii) the primary and secondary bile acid (BAs) pathways [12], and (iv) the *vagus nerve*. Microbiota-derived molecules act either by functionally interacting with other endocrine hormones (i.e., ghrelin, leptin, glucagon-like peptide 1, and peptide YY) or with the immune system (altering, for example, the levels of circulating cytokines) [4,13]. Strikingly, the maternal gut microbiota drives the early development of the immune system [14]. In addition, the gut microbiota can stimulate the parasympathetic nervous system, thereby impacting glucose homeostasis and other metabolic processes linked to the production of microbiota-generated metabolites promoting metabolic benefits, for example, in promoting body weight and glucose control (demonstrated in animal models) [15]. Strikingly, despite the presence of the brain-blood barrier, these molecules can reach the brain thanks to the *brain-gut-microbiota axis*: a bidirectional communication system enabling gut microbes to communicate with the brain (and the brain with the gut), and it can have a profound effect brain physiological state [16]. 

### 1.2. Fecal Microbiota Transplantation: Advantages, Preparation and Delivery Methods

Different strategies exist to modulate an altered dysbiotic gut microbiota composition aiming at a more physiological profile. These therapies can either (i) target/eliminate specific pathogenic strains using antibiotics (“*antimicrobial therapy*”) or phage (“*page therapy*”), or (ii) administrate specific live microbe (in the form of “*probiotics*”) or (iii) transfer entire microbiota communities (“*fecal microbiota transplantation*”) [17]. *A**ntimicrobial therapy* was the first that came into place and it aimed to directly target the microbes that necessitated being controlled/restricted by means of antibiotics. However, it quickly lost therapeutic interest because of the lack of new anti-bacterial agents and the spread of antimicrobial resistance [18]. The successive *phage therapy* exploited bacteria-specific viruses (phages) to combat populations of pathogenic bacteria [18,19,20], for example, against the multi-drug-resistance *Staphylococcus aureus* [21]. In comparison to chemical antibiotics used in antimicrobial therapy, the phages limit antibiotic resistance because directly kill bacteria [22], and they exert a minimal disruption of normal flora [23] thanks to their specific host selectivity [19]. However, the umbrella of phages, thus targeted bacteria, that meet the criteria as therapeutics is limited [24] and the general perception of phages as “viruses that lead to human diseases” might limit its acceptance and exploitation in therapy. Beyond the antimicrobial and phage therapies, it is also possible to regulate gut microbiota composition utilizing the administration of specific living bacteria in the form of *probiotics*. Probiotics are administered as a supplement and have shown to have enormous beneficial effects on a wide range of gastrointestinal diseases and brain pathologies [17,25]. However, probiotics still come with limitations. Indeed, despite many promising bacterial families that could have beneficial effects, only a few are available in the form of probiotics. Meanwhile, very different from other techniques aimed to alter/change the gut microbiota, the *fecal microbiota transplantation* (FMT) technique transfers the entire, complete, stable fecal microbial community of gut micro-organisms contained in the feces from healthy donors to people with a particular disease (associated with altered microbiota) with the final goal to restore dysbiosis and tackle disease’s symptoms [26,27,28,29,30,31]. In comparison to probiotics that could transfer a few microbial species into the host gut, FMT has the greater advantage of transferring the entire gut micro-ecology as a proper “organ” and, differently from other organ transplantations, it is highly safe and does not trigger an immune response or rejection [29].

Fecal stools derived from selected donors need to be processed and prepared before being transplanted into the recipient. How is FMT prepared? The detailed method varies across the different studies. In general donor stools (~100–150 g) are collected and a sterile saline solution (NaCl, 0.9%) is added for a preliminary homogenization to get a feces slurry employing a speed blender (Figure 1) [28,32,33]. Then, larger particles, fibers, and undigested food are removed by filtration using a metal sieve, and the homogeneous liquid *fresh* fecal sample can be transferred in sterile syringes and ready for FMT within less than 6 h after the emission. The fresh fecal preparation was the first to be used for *C. difficile* infections [34]. Alternatively, the preparation can be further processed with multiple steps of filtration where the diameters of the filters keep decreasing (from 2 to 0.1 mm), cry-protected in glycerol (10%), frozen, and kept at −80 °C for later FMTs; prior FMT, the frozen slurry has to be thawed at 4 °C overnight and reconstituted with normal saline (Figure 1) [33,35].

To prepare FMT capsules, freeze-drying protectant glycerol (20%) is added to the initial fecal preparation, and centrifuged (400× *g*); then the supernatant is discarded and it is centrifuged again at high speed (10,000× *g*), the sediment is incorporated into enteric-soluble capsules and stored at −80 °C [35]. Alternatively, the temperature of the recovered sediment is lowered slowly to −80 °C, and afterward lyophilized (vacuum dried) to obtain fecal powder. ~0.9 g of powered fecal preparation are inserted in each enteric-soluble capsule, and stored at −80 °C for later use (Figure 1) [35,36,37,38].

Recently, the multiple steps of microfiltration, centrifugation, and suspension have been carried out via an automatic purification system referred to as washed microbiota preparation (WMP) that aims to deliver a precise dose of the enriched microbiota instead of using the weight of the stool sample [39] (Figure 1). WMP is shown to wash out a higher type and amount of viruses and to reduce three times the adverse effects [39]. In order to standardize WMP methodology, a consensus was reached recently [40].

Once the stool material has been processed and is ready, it can be administered for FMT by using a variety of delivery methods broadly classified into upper (nasogastric, nasoduodenal, and nasojejunal tubes, and capsules) and lower gastrointestinal routes (enema, colonoscopy) (Figure 2). Traditional *enema**,* also known as a clyster, was the simplest first method to be exploited and it consists of an injection of the stool preparation into the lower bowel by way of the rectum. FMT via retention enema in hospitalized patients with severe or complicated CDI is shown to be effective and well-tolerated and effective at resolving symptoms of CDI in patients with multiple underlying co-morbidities [41]. Successively, FMT was applied by *colonoscopy* (Figure 2) where the physician infuse the liquefied donor’s stool into the end of the small intestine or the beginning of the colon by means of a colonoscope [42] (for a video of the procedure check the research article [34]). With respect to an enema, the advantage is that the procedure is visually inspected and guided. FMT via colonoscopy was reported to be an effective treatment in severe or complicated *C. difficile* infection, though this protocol requires a trained endoscopist, it is costly, and it carries procedural risks; it cannot be performed in subjects with colon inflammation. Different from the classic enema or colonoscopy, colonic *transendoscopic enteral tubing* (TET) involves the placement of a tube through the anus into the cecum for whole-colon administration (Figure 2). The tube could be maintained for repeated FMTs. TET showed to be effective and less psychologically challenging for patients [43,44]. When colonic TET is used to deliver a washed microbiota preparation (WMP), it is referred to as *washed microbiota transplantation* (WMT) (Figure 1 and Figure 2). 

Among the upper gastrointestinal routes, FMT can be delivered by means of a nasal tube (inserted through the nose) and reaching to deliver the fecal transplant either in the stomach (nasogastric tube), in the duodenum (nasoduodenal tube), or in the jejunum (nasojejunal tubes) [45] (Figure 2). The most recent delivery method occurs via oral administration of encapsulated FMT, which is more tolerated by the patients and was shown to be safe and effective [46,47] (Figure 2). If the fecal stool comes from the same patient who will receive it, then it is referred to as “*autologous FMT*”; however, if it is from another person then it is referred to as “*allogenic* or *heterologous* FMT”.

## 2. Fecal Microbiota Transplantation in Human Diseases

### 2.1. Fecal Microbiota Transplantation: First Use for C. difficile Infections

FMT finds its roots further back in time than we may imagine. In the 4th century during the Dong-jin dynasty in China, the traditional Chinese medicine doctor Ge Hong used a human fecal suspension (known as “yellow soup”; administered from the mouth) for the treatment of severe diarrhea [48]. Over the last 50 years and still to this day, FMT is widely recommended under clinical guidelines only to efficiently resolve both *primary* [49] and *recurrent* (with each recurrence increasing the probability of a successive recurrence) refractory infections caused by the pathogen *C. difficile* both in adult [29], elderly (~82 years old), and debilitated patients [50] Contrarily to what previously believed, the stool donor body mass index does not affect the recipient body weight [51]. One factor that might decrease the efficacy of FMT in *C. difficile* infections is the virome dysbiosis observed in the recipients showing an abundance of bacteriophage *Caudovirales* [52]. Studies showed that: (i) heterologous FMT is more effective than autologous FMT [53,54]; (ii) the use of frozen FMT is not inferior compared to fresh stools in the clinical outcome [55,56,57,58,59], but the lyophilized form is slightly less effective than the fresh one [59], and (iii). FMT often requires only a single administration, it results in *C. difficile* eradication without the need for complete microbiota engraftment [54], and in a significant and long-term change in the gut microbiota composition in the patient [60,61]. FMT was even described to be superior to standard fidaxomicin and vancomycin, first-choice drugs to treat CDI [62]. Nowadays, clinical trials exploiting FMT for *C. difficile* are even performed at home [63].

To shed light on how FMT reconstitutes the gut microbiota, Kumar and colleagues (2016) analyzed the colonization potential of the donor, recipient, and recipient post-FMT using human fecal transplantation in gnotobiotic mice. Microbiome analysis showed that members of the family *Bacteriodaceae* and *Lachnospiraceae* were highly represented in the donor stools, but not in the recipients’ pre-FMT who contained *Enterobacteriaceae*, *Lactobacillaceae*, *Enterococcaceae,* and an abnormally higher proportion of *Clostridiales* (including *C. difficile*). The analysis of microbiota profiles in gnotobiotic mice transplanted with fecal stools coming from patients who received FMT after three days revealed increases in the relative abundance of *Bacteriodaceae* and *Lachnospiraceae* to levels similar to the donor, but this relative abundance dropped to 7% of that of the donor in gnotobiotic mice transplanted with using stools coming from patients who received FMT in the past 2–4 weeks. These data proposed that after FMT, the commensal microbes, *Bacteriodaceae* and *Lachnospiraceae,* at early times post-FMT start colonizing the receiver’s gut and compete with non-commensal *Clostridiales* to occupy niche space [64].

Despite being firstly and mainly used to treat infections caused by *C. difficile*, FMT has been exploited in a number of human conditions [26,65]. In this review, we will summarize all the clinical trials on several human pathologies exploiting FMT strategies (Figure 3), and we will propose how FMT will eventually provide in the nearly future a new therapeutic tool aiming to ameliorate clinical symptoms to be used in the first place rather than being used as parallel therapy when GI dysfunctions are also involved. We will highlight that more research is needed considering that “understanding of the efficacy of FMT in pathologies other than C. difficile infections is still very much in its infancy [47]”.

### 2.2. Fecal Microbiota Transplantation in Gastrointestinal Diseases

#### 2.2.1. Inflammatory Bowel Disease

Among human conditions affecting the GI tract, *inflammatory bowel disease* (IBD) is a complex inflammatory and chronic disease characterized by immune dysregulation that ultimately results in immune-mediated damage to the alimentary tract. Among IBD types, Crohn’s disease and ulcerative colitis are the principal ones: the first one affects the mouth, the esophagus, the stomach, the small and large intestines, and the anus, whereas the second one primarily affects the colon and the rectum [66,67]. Symptoms include abdominal pain, diarrhea, rectal bleeding, and anemia. Current therapeutic strategies mostly rely on the direct targeting of the immune response. Nevertheless, these therapies present a high cost together with an increased risk of adverse events and infections [68,69]. Dysbiosis in the gut microbiota is considered to be a key regulatory event in IBD development [70,71], thus FMT represents a possible therapeutic strategy [72].

The first clinical trials took into account patients with the two main forms of IBD; few patients showed clinical remissions associated with a high donor gut microbiota richness [73].

The next clinical trials focused on only one form of IBD. In a first placebo-controlled randomized trial focused on patients with active *ulcerative colitis* (without infectious diarrhea), Moayyedi and colleagues (2015) administered (50 mL) via retention enema once a week for six weeks. The trial revealed no adverse events following FMT showing its safety, and 24% of patients showed remission from ulcerative colitis [74]. In a second double-blind, randomized, placebo-controlled trial, patients with active ulcerative colitis were treated by FMT colonoscopic infusion, followed by enemas five days per week for eight weeks with a resulting 27% of remission but 78% of adverse events in patients. Analysis of the ribosomal 16S RNA revealed an increase and persistent microbial diversity after FMT; in particular, the strain *Fusobacterium* spp. was associated with a lack of remission from ulcerative colitis [75]. It is very clear to notice the very long treatment period used in the first two clinical trials described, a long period that surely creates discomfort in the patient receiving FMT, but it also implies an extensive effort from the medical team. In this view, a third trial reduced drastically the duration of FMT treatment via a nasoduodenal tube to only the start of the study, and three weeks later, showed no statistically significant difference in the remission rate between patients who received FMT, showing that ~30.4% of patients who received allogenic FMT and 20.0% who received autologous FMT obtained clinical remission. However, no differences were found when comparing the two groups [76]. In a fourth trial, Costello and colleagues (2019) enrolled people with mild to moderate ulcerative colitis and further limited the duration of FMT to only seven days applied via colonoscopy followed by two enemas. Two months after FMT, 32% of the patients showed remission, and this data represented a higher remission rate in a shorter time than in the former studies [77]. Instead, only one single FMT by colonoscopic administration resulted ineffective in achieving clinical response in a separate trial [78], but if the single FMT administration is performed by using a high-diversity FMT (pooling two donors) is effective in achieving clinical remission (in 35% of the patients) and increased microbial diversity [79]. Taken together, both the number of FMTs and donor diversity play a major role in the final clinical outcome.

The described four randomized controlled trials included a total of 277 patients and showed variable remission rates from ulcerative colitis. Nevertheless, all four trials exploited multiple endoscopic or enema-based treatments, raising obvious concerns regarding their long-term feasibility. Recently, Crothers and co-workers (2021) undertook a different approach using a first FMT induction by colonoscopy, then followed by 12 weeks of daily oral administration of frozen encapsulated FMT (containing ~ 0.5 g of stool). The treatment was shown to be well-tolerated and safe reporting only a few adverse events and no infectious complications, thereby opening a new path for FMT administration likely to be accepted by some patients as a therapeutic alternative to treat ulcerative colitis. Importantly, the gut microbiota of the receivers correlated with those of the donors for up to 20 weeks. Despite the demonstrated safety of oral FMT treatment, the amount of data collected in this study was not enough to evaluate the effects of oral FMT treatment on the clinical outcome [46]. Moreover, another problem concerned that one of the frozen oral FMT capsules could undergo different freeze-thaw cycles during its transport to home and home freezer conditions, thereby calling for more temperature-stable formulations and further clinical trials addressing the efficacy and the optimal dose. This further refinement of oral capsule stability and preparation came very recently with the clinical trial performed by Haifer and colleagues (2022) who exploited lyophilized oral FMT capsules for ulcerative colitis [36]. After a two-week treatment with antibiotics, the patients received oral lyophilized FMT or placebo capsules for eight weeks. At the end of the treatment, the patients in the FMT group reported higher clinical, endoscopic, and histologic remission rates in comparison to the placebo group. These FMT patient responders were divided into two groups to either continue or withdraw FMT (mimicking the classic washout period) for a further 48 weeks. At the end of the second FMT round, only patients that continued the treatment were still in clinical remission, while the ones who had FMT withdrawn lost the remission [36], pointing to the importance of continuous FMT treatment to achieve long-term and stable clinical remission.

Given the number of clinical trials performed and the variability in the remission rate, an important aspect to be investigated regarding FMT in people with ulcerative colitis concerns the specific microbiota profile achieved after FMT and associated with a clinical response, together with the level of engraftment. To tackle this issue, a first small trial in five patients found out that after a single FMT by colonoscopy, the donor similarity index was 40–50% in 60% of the patients, and it correlated with clinical remission [80]. Paramsothy and colleagues (2019) analyzed the fecal samples collected before and after intensive FMT treatment (five days per week, for eight weeks) and found out that FMT did increase the microbial diversity, but this increase was higher in patients who achieved remission and was associated with an enrichment of *Eubacterium hallii* and *Roseburia inulivorans* and increased levels of SCFAs compared with patients who did not achieve remission [81]. Instead, patients who did not show remission had an enrichment of *Fusobacterium gonidiaformans*, *Sutterella wadsworthensis* and *Escherichia* species, and lipopolysaccharide (LPS) level. Interestingly, when correlating donor microbiota profiles with remission, the presence of *Bacteroides* in the donor stool was associated with FMT response, while *Streptococcus* species were associated with a lack of response [81]. The long-term sustained remission is associated with overall increased butyrate production and levels of butyrate producers [82]. These pieces of evidence again point out the pivotal importance of the donor microbiota profile and the number of transplants choice when designing an FMT intervention for ulcerative colitis.

So far, the described clinical trials were performed in adults with ulcerative colitis, but recently interest in FMT strategies come into the spotlight also for *pediatric patients*. Following a few trials with a low number of pediatric patients yielding conflicting results [83,84,85,86,87], Pai and co-workers (2021) performed the first randomized clinical trial in pediatric patients (aged 4 to 17 years) with active ulcerative colitis. Ninety-two percent of the patients in the FMT arm achieved improvement in pediatric ulcerative colitis activity index (compared to the 50% of the placebo arm) at week 6, and 75% still showed clinical response one year after the transplant. The bacterial taxa *Alistipes* spp. and *Escherichia* spp. were associated with the clinical outcome [88].

IBD can be complicated by infections caused by cytomegalovirus (CMV), indeed the virus passes from the latent to the active form triggered by immunosuppressive drugs and leads to CMV colitis [89,90] increasing the risk for colectomy in patients with ulcerative colitis [91]. If patients are found positive for CMV then discontinue the therapy with immunosuppressive drugs and start therapy with the anti-viral drug ganciclovir [92] which has high efficacy but presents serious adverse effects [93]. Given the similarities between CMV and *C. difficile* infections in IBD, and the effectiveness of FMT in *C. difficile* eradication in IBD, Karolewska-Bochenek, and co-workers (2021) explored FMT in eight children with ulcerative colitis and CMV re-infection. The children received FMT via nasogastric tube on five consecutive days every two weeks and were assessed for clinical remission after six weeks. Three out of eight children achieved clinical remission, and no adverse effects were recorded during or after the treatment [94]. Despite efficacy not being achieved in the majority of the patients, it still provided a new and promising therapeutic option for CMV colitis.

Another major form of IBD is Crohn’s disease, and it is associated with gut microbiota dysbiosis with reduced diversity of bacterial phyla including an abundance of bacterial families such as *Veillonellaceae*, *Enterobacteriaceae*, *Pasteurellaceae*, *Fusobacteriaceae* along with decreases in *Erysipelotrichales*, *Bacteroidales* and *Clostridiales* correlated to disease severity [95]. Another study found that patients with Chron’s disease have lower levels of *Bacteroides*, *Eubacterium*, *Faecalibacterium*, and *Roseburia*, and higher levels of *Clostridium*, *Cronobacter*, *Fusobacterium*, and *Streptococcus* [96]. This piece of knowledge together with the fact that probiotics have some efficacy, but still their repertoire is limited prompted to opt for FMT [72].

Some clinical trials evaluated the efficacy of a single FMT. He and co-workers (2017) evaluated the efficacy and safety of multiple fresh FMTs (an initial FMT followed by repeated FMTs every three months) in 25 people with Crohn’s disease also suffering from the intra-abdominal inflammatory mass. More than half of the patients showed clinical response and remission three months after the first FMT, with this percentage decreasing at 12 and 18 months, suggesting that despite relieving the clinical symptoms in the short term, FMT fails to induce a long-lasting clinical effect [97]. Vaughn and colleagues (2016) investigated in a small group (nineteen subjects) the effect of a single FMT not only on clinical remission but also on mucosal inflammation by analyzing mucosal T-cell phenotypes and inflammatory parameters. FMT resulted in remission in roughly half of the patients, an increase in gut microbiota diversity, and a number of regulatory T-cells [98]. Even if FMT single administration seems promising in the short term, it fails to maintain clinical remission in the long term. The question is when, timewise, the second FMT should be performed to maintain long-term the positive effects of the first dose. Li and colleagues (2019) established that the critical time window is four months, meaning that a patient with Crohn’s disease who underwent clinical remission with a first FMT is more likely to maintain that remission if receives a second FMT within four months from the first one [99].

Different parameters could play a role in FMT outcomes, in general, and in particular in Crohn’s disease. For example, one would question whether the way how FMT is prepared, or the different methods of delivery could influence FMT outcomes. To answer the first question, Wang and colleagues (2018) evaluated the risk factors of adverse events based on different methods of preparation of fecal samples for FMT, finding that the manual methods were associated with a higher probability (~22%) for the patients undergoing adverse events than the automatic methods (~9%), despite the method *per se* (manual or automatic) did not affect FMT clinical efficacy [100]. To answer the second question, a recent study compared clinical remission rate and adverse events of FMT delivered either by gastroscopy or by colonoscopy, finding no differences; moreover, as shown by other studies, FMT increased gut microbiota diversity in the patients [96]. Can FMT maintain the clinical remission achieved after drug therapy? In a study performed by Sokol and colleagues (2020) patients with Crohn’s disease were enrolled and clinical remission was achieved by drug therapy by corticosteroids; then, corticosteroids were reduced, and patients received either FMT or sham transplantation via colonoscopy. The FMT group showed a higher remission rate than the sham group, and the absence of engraftment was associated with flare [101].

Surely, clinical trials investigated FMT strategies to achieve clinical remission in patients with IBD and also recurrent *C. difficile* infections, especially in the light that FMT was firstly exploited at its origin to eradicate refractory *C. difficile* infections and that IBD is associated with a higher prevalence of infections by *C. difficile* [102,103,104] the latter known to aggravate IBD pathology per se [105]. In Europe, a first pilot study in one patient with IBD and *C. difficile* infection showed clinical remission one year after FMT [106]. Ianiro and colleagues (2021) took it further and investigated it in the first large European study on the topic. They enrolled patients with IBD (Crohn’s disease and ulcerative colitis) who saw their clinical condition worsen after infection by *C. difficile*. Patients received FMT via colonoscopy based on IBD severity and the conditions: each patient received at least one FMT, but hospitalized patients received sequential FMT, and patients with pseudomembranous colitis underwent FMT until its disappearance. Eight weeks after FMT, 94% of the patients were negative for *C. difficile*, and mostly reported improvement in clinical conditions [107].

The clinical trials reviewed so far for FMT strategies in Crohn’s disease focused on adult patients. One double-blind, randomized, placebo-controlled pilot study in progress (NCT03378167) will explore the feasibility, adverse events, clinical efficacy, and change in the microbiome of FMT in 45 pediatric patients (from 3 to 17 years of age). The FMT group will undergo a colonoscopic infusion followed by oral capsules two times per week for six weeks. Results are expected by December 2022 [108].

Taken together, a number of clinical trials focused on the two major forms of IBD, namely, ulcerative colitis and Crohn’s disease, clearly showed the safety and efficacy of FMT in achieving clinical remission in adult patients. FMT showed to also be effective in maintaining clinic remission after being achieved via drug therapy. Importantly, apart from a few studies, many clinical trials reviewed here explored FMT strategies independently from concomitant recurrent *C. difficile* infections.

#### 2.2.2. Irritable Bowel Syndrome

Differently from IBD which involves inflammation and destruction of the bowel wall, irritable bowel syndrome (IBS) does not involve inflammation and it causes stomach cramps, bloating, diarrhea and constipation. Gut microbiota composition in people with IBS differs from healthy subjects [109] and it plays a role in its pathophysiology. Nine randomized controlled trials investigated FMT interventions in IBS with contrasting results. 

On one hand, six different clinical trials found positive effects of FMT on IBS symptoms: (i) in a small clinical trial with just ten patients enrolled, Mizuno and co-workers (2017) reported an improvement in six patients four weeks after FMT; interestingly, the authors found that the patients that reported an improvement received FMT from a donor with a higher content of *Bifidobacterium* than in ineffective donors indicating that *Bifidobacterium*-rich fecal donor might be a predictor for a successful FMT [110]. (ii) Johnsen and colleagues (2018) observed symptoms relief in 65% of patients three months after being treated with FMT via colonoscopy [111] together with a shift of the microbial profile toward the donor profile following FMT including an increased alpha and beta diversities [112]. (iii) Mazzawi and colleagues (2018) reported an improvement in symptoms and quality of life in patients with diarrhea-predominant IBS who were administered fresh fecal stools via a gastroscope [113], and (in a successive analysis) increased SCFAs and alignment with donor gut microbiota composition in IBS patients after FMT [114]. (iv) Lahtinen and co-workers (2020) reported a transient improvement in IBS symptoms in patients receiving either fecal material derived from a healthy donor (allogenic transplant) or from themself (autologous transplant); the patients receiving an allogenic transplant recorded a decrease in the depression score [115]. (v) in order to standardize donor diversity, El-Salhy and colleagues (2020) treated patients with IBD via gastroscope FMT using fecal samples obtained only from one healthy, well-characterized donor, and found that patients receiving 30 g (~77%) and 60 of FMT (~90%) reported improvements in fatigue and quality of life together with a change in bacterial microbiota profiles [116], change in fecal SCFAs [117], without sex difference in the response to FMT (i.e., improvement in symptoms, microbiota profiles, level of SCFAs), even if the response rate was significantly higher in females than in males [118]. (vi) in a recent clinical trial, Holvoet and colleagues (2021) recruited people with refractory IBS (meaning that patients failed to report improvement in symptoms in at least three conventional therapies) with predominant bloating and treated with one FMT dose via nasojejunal administration; after one year of FMT, 56% of patients reported improvement of IBS symptoms and in the quality of life; moreover, the authors observed that (before FMT) the gut microbiome of the respondents had higher diversity (no specific taxa involved) than the non-respondent, suggesting this could be used as a marker to predict FMT effects [45]. On one hand, these clinical trials reported an improvement of IBS symptoms together with changes in microbiota profiles and SCFAs after FMT in IBS patients, on the other hand, the other three clinical trials found no major effects. (vii) Halkjær and co-workers (2018) treated people with moderate-to-severe IBS with FMT capsules for 12 days and observed an improvement in IBS symptoms after three months together with an increase in gut microbiota diversity when comparing the stools collected before and after FMT; disappointingly, six months after the patients in the placebo group reported better symptoms relief than in comparison to the FMT group, suggesting that modifying gut microbiota composition might not be enough to ameliorate the symptoms [119]. (viii) Aroniadis and co-workers (2019) recruited people with diarrhea-predominant IBS and treated them with over 25 FMT capsules/day for three days (each capsule contained 0.38 g of minimally processed donor stool) but no improvement in symptoms was reported at three months in comparison with the placebo [120]. (ix) Holster and co-workers (2019) compared, in a small group of patients, FMT (via colonoscopy) using fecal material derived from a healthy donor (allogenic transplant) or from their own one (autologous transplant) finding no major differences in symptoms improvement between the two groups, despite an improvement in comparison to the baseline in the allogenic group [121]; the same research group found altered interactions between the gut microbiota and its metabolites [122] and an analysis of the RNA isolated from colonic biopsies before and after FMT revealed an activation of immune response-related genes in patients treated with allogenic FMT, and an activation of metabolism-related genes in those ones treated with autologous FMT [123].

The contrasting results in the described clinical trials might be due to the donor variability (in fact, certain donor microbiota profiles were shown to be predictive of a positive response in IBS patients), but also the dose of the transplant might be important for the successful outcome. El-Salhy and co-workers (2019) investigated whether a high transplant dose and/or repeating FMT were required to obtain a positive response, finding that it actually might be the case: in fact, they selected patients with IBS who did not respond to a 30 g FMT and treated them with 60 g transplant into the duodenum via a gastroscope. Most of the patients who were unresponsive to the first dose now responded to the higher dose showing improvements in clinical symptoms, fatigue, quality of life, and dysbiosis index [124]. Thereby, the ideal FMT therapy for each patient with IBS would better consist of an attentive selection of the donor together with a personalized dosage depending on the response to FMT.

#### 2.2.3. Other Gastrointestinal Diseases

The small intestine is usually populated under physiological conditions by a lower number of bacteria in comparison with the colon. However an unusual excessive bacterial growth gives rise to the so-called *intestinal bacterial overgrowth* (SIBO), a disorder characterized by nausea, vomiting, bloating, diarrhea, malnutrition (in children), weight loss, fatigue, weakness, and steatorrhea (presence of fat in the stools) [125]. SIBO is frequent in people with IBS [126]. To tackle SIBO, either antibiotics [127] or probiotics [128] had been exploited but still with limited efficacy. Given FMT’s positive effects on main GI diseases in general and on IBS in particular (as mentioned earlier, people with SIBO might experience IBS too [126]), Xu and colleagues (2021) explored for the first time the clinical efficacy of FMT for treating SIBO. Patients with moderate to severe SIBO (who did not get any antibiotic treatment in the two previous months) received oral FMT capsules or placebo once a week for four weeks and were followed up for six months. In order to measure the level of bacteria, the authors measured the gas released by means of the lactulose hydrogen breath test. Besides no side effects observed, oral FMT resulted in an improvement of a variety of GI symptoms (abdominal pain, reflux, indigestion, diarrhea, and constipation) and a reduced gas increase compared to the placebo. Ribosomal 16S RNA analysis showed that the donors had higher microbiota diversity than the patients, and FMT resulted in an alteration of *Bacteroides* abundance (at the genus level).

### 2.3. Fecal Microbiota Transplantation in Liver Diseases

Among liver diseases, *primary sclerosing cholangitis* is a cholestatic liver disease with unmet clinical therapy and is associated with dysfunction in gut microbiota. Allegretti and co-workers (2019) recruited ten patients (who also suffer from concurrent IBS) who underwent a single FMT by colonoscopy. Thirty percent of the patients experienced a major decrease in the level of alkaline phosphatase (an indirect measurement of liver damage) that correlated with the engraftment of donor taxa and bacterial diversity post-FMT [129].

*Recurrent hepatic encephalopathy* is a complication of hepatic cirrhosis (not associated with alcohol intake), it results from liver failure and gut-liver-brain axis impairment and it can lead to consciousness impairment and coma [130]. Despite the use of antibiotics, rifaximin is effective in reducing the risk of another episode of hepatic encephalopathy [131], it still leads to significant mortality and FMT has been recently investigated as a possible new therapeutic tool. A first trial described that a single FMT delivered by enema reduced hospitalization and improved cognition and dysbiosis in people with hepatic encephalopathy in the short term [132]. A second trial instead focused mostly on the long-term effects (up to 12–15 months) upon a single FMT administration, still by enema. FMT reduced the hospitalization rate and improved cognitive function with respect to the control group up to one year from FMT treatment [133]. The study also analyzed the microbiota composition finding a lower representation of *Lachnospiraceae* and *Ruminococcaceae* in patients versus controls that did not change after FMT; instead, in the long term, FMT increased the abundance of *Burkholderiaceae* and decreased that of *Acidaminoccocaceae* [132]. A follow-up study used capsules, instead of enema, to deliver FMT derived from a single donor enriched in *Lachnospiraceae* and *Ruminococcaceae*, namely, the bacteria found to be reduced in patients, and found improved duodenal mucosal diversity and dysbiosis [38]. FMT by enema enriched in *Lachnospiraceae* and *Ruminococcaceae* showed to be effective in restoring antibiotic-associated disruption of gut microbiota diversity and function in people with cirrhosis

The latest described clinical trials focused on FMT interventions in subjects with liver disease not associated with alcohol intake. Nevertheless, cirrhosis can be complicated by *alcohol use disorder* (AUD) which represents a major cause of mortality worldwide [134], thereby therapies focused on alcohol intake control are relevant. Given that (i) AUD impacts the gut microbiota composition (even before disease development and it worsens over time) [135], and that (ii) the microbiota may play a role in addictive behaviors [136], Bajaj and colleagues (2021) investigated whether FMT could reduce alcohol craving in a small group of subjects with cirrhosis and AUD. The patients received 27 g of stool or placebo by enema. The selected donor stools contained higher levels of *Lachnospiraceae* and *Ruminococcaceae* that were underrepresented in the patients. In comparison to the placebo group, subjects who received FMT exhibited a higher diversity in their microbiota (higher relative abundance of *Odoribacter*, *Bilophila*, *Alistipes,* and *Roseburia*), and higher plasma levels of SCFAs. Importantly, FMT reduced alcohol craving and improved the quality of life and cognition. The genera *Bilophila* and *Ruminococcus* were associated with reduced alcohol craving [137]. Taken together, the described clinical trials focused on FMT interventions for liver cirrhosis not only suggest the potential benefits of microbiota-based interventions in liver diseases (associated or not with alcohol intake). Moreover, it emphasized the importance of donor selection owning a specific microbiota profile, in this case, more enriched in bacteria lacking in the patients), for a more successful clinical outcome.

*Nonalcoholic fatty liver disease* (NAFLD) is an obesity-related disorder characterized by an excessive fat build-up in the liver in people who drink little to no alcohol. It may cause fatigue, pain in the upper right abdomen, abdominal swelling, and an enlarged spleen. Despite the known gut microbiota dysbiosis in people with NAFLD, so far FMT failed to improve insulin resistance, but it did reduce the small intestinal permeability [138].

Liver diseases can be caused also by viral pathogens. For example, the hepatitis B virus (HBV) is an important cause of *chronic hepatitis B* liver disease, a worldwide public health challenge leading to cirrhosis, liver failure, or hepatocellular carcinoma. During chronic HBV infection, the immune response transits from an active to an inactive state which fails to clear the virus [139]. Pieces of evidence in the literature show that (i) the human gut microbiota profile is altered in HBV (decreased *Bifidobacteriaceae*/*Enterobacteriaceae* ratio, low levels of *Bifidobacteria* and *Lactobacillus*, high levels of *Enterococcus* and *Enterobacteriaceae*) [140], and (ii) the gut microbiota promotes liver immunity resulting in HBV clearance in a mouse model [141], thereby suggesting that the gut microbiota might help in the immune response. In order to explore FMT-based strategies in HBV, Chauhan and colleagues (2021) performed a pilot study evaluating the loss of hepatitis B surface antigen (HBsAg) with FMT. A group of 14 patients in antiviral treatment underwent six cycles of FMT via nasoduodenal tube every four weeks, while another group of 15 patients under antiviral treatment was taken as control. FMT was well tolerated, and the patients in the FMT group had HBsAg clearance in comparison to the control [142]. Despite a larger cohort study being needed, this study showed FMT to be effective in terms of viral suppression and HBsAg clearance in patients with HBV.

### 2.4. Fecal Microbiota Transplantation in Obesity and Metabolic Disorders

Nowadays, obesity and metabolic syndromes represent a major health epidemic and challenge the prevention of chronic diseases [143]. Despite having complex multifactorial origins (including genetic, behavioral, and environmental ones), over the last 30 years in many countries, obesity has been also driven by extensive urbanization, sedentary lifestyle, and a nutritional transition to processed foods. The current medical strategies still have limited efficacy and tolerance (for example, liraglutide [144,145]) and high cost [146,147,148], and lifestyle changes and antidiabetic agents are not able to reduce morbidity and mortality rates [149]. Critically, taking into consideration the actual trend, time trend forecasts predicted that by 2030, 51% of the population will be obese with a consequent increase in healthcare costs [150]. Given the continued lack of progress in providing newer and effective therapies and the known influence of the gut microbiome on obesity and metabolic syndromes [151,152], growing attention and interest had been directed toward FMT [32,153,154].

Based on pre-clinical and pioneering experiments showing that the obese and lean phenotypes could be transferred in rodents by using the fecal microbiota of human donors [155], different successive clinical studies emerged. Firstly, three clinical trials investigated the effect of FMT on lean healthy donors to people with metabolic syndrome showing improvements: Vrieze and colleagues (2012) reported an increased insulin sensitivity in male participants with metabolic syndrome who underwent a six weeks infusion of intestinal microbiota from lean donors [156]; Kootte and colleagues (2017) recapitulated the latter findings on increased insulin resistance and added that it is dependent on changes in intestinal microbiota following FMT [157]; finally, de Groot and co-workers (2020) showed that when obese and insulin-resistant male subjects were transplanted using feces derived from donors with metabolic syndrome showed a decreased insulin sensitivity compared with subjects transplanted with feces derived from normal donors, thereby showing a causal link between insulin-sensitivity and microbiota in metabolic syndrome [158].

Although interesting, the described three clinical trials presented limitations because (i) the improvements were short-term, (ii) the studies focused only on male participants, and (iii) they used invasive FMT delivery. Recently, independent clinical trials investigated both male and female participants, FMT by using oral capsules (instead of the classic invasive delivery methods). Different studies explored FMT in adult participants [37,47,159], while others focused on adolescent patients [160,161]. In the first study, Yu and co-workers (2020) evaluated in a small group of adult patients with obesity and mild-moderate insulin resistance (~70% females) the effect of a weekly administration for six weeks of FMT capsules containing fecal microbiota derived from a healthy lean donor.; despite successful variable engraftment into the recipient’s microflora, the authors did not observe any significant metabolic improvements in terms of either insulin sensitivity or body composition [159]. The study could have been influenced by the inclusion of participants with relatively mild insulin resistance, but Allegretti and colleagues (2020) obtained the same negative results akin Yu and co-workers (donor microbiota engraftment but no metabolic improvements) in a small group of obese metabolically uncompromised patients (without a diagnosis of diabetes or metabolic syndrome) [37], pointing to the fact that the lack of effect on the metabolic outcomes cannot be related to the presence of other metabolic conditions. Instead, the negative results could be related to the small group size and, especially, the lack of dietary intervention, the latter being the key to a successive study. In fact, Mocanu and colleagues (2021) evaluated in a phase two trial the effect of a single oral encapsulated FMT (aiming to alter the recipient microbial flora) combined with adjunctive daily fiber supplementation (aiming to enhance and/or maintain FMT-derived changes) in a sample of adult males and females with severe obesity and metabolic syndrome. FMT and fiber supplements were well-tolerated among participants. As a primary outcome, patients treated with FMT plus low-fermentable fiber showed increased insulin sensitivity between baseline and after six weeks of treatment, while no differences were reported in the group treated with FMT plus high-fermentable fiber or fiber alone. The increased insulin sensitivity was driven by an improvement in serum insulin levels. At the level of microbial ecology, the treatment with FMT plus low-fermentable fiber increased alpha (up to 12 weeks) and beta (up to six weeks) diversities, inducing, in particular, an increase in the relative amount of *Phascolarcobacterium*, *Christensenellaceae*, *Bacteroides* and *Akkermansia muciniphila*, and a decrease in *Dialister* and *Ruminococcus torques*. Interestingly, *Phascolarctobacterium*, *Bacteroides stercoris,* and *Bacteroides caccae* presence in the baseline microbial taxa composition was a predictor of the improved insulin sensitivity after FMT, suggesting that these taxa could be used as treatment given their responsiveness to microbial biotherapeutic intervention. Lastly, the authors showed that only the group treated with oral FMT plus low-fermented fiber showed a significant increase in bacterial richness and a shift in microbial composition toward the donor composition [47]. Thus, this recent study showed that a single oral FMT administration coupled with daily low-fermentable fiber supplementation can efficiently improve insulin sensitivity and microbial diversity in people with severe obesity and metabolic syndrome providing a new non-invasive tool for microbial biotherapeutic strategies.

As far as FMT clinical trials in adolescents with obesity are concerned, Wilson and co-workers (2021) evaluated strain engraftment after the administration of multi-donor FMT capsules (containing the fecal microbiota of four sex-matched lean donors). Twenty-eight capsules were administered over two consecutive days. The authors reported a general efficacy in the multi-donor FMT to alter the structure and the function of the gut microbiome, but in particular, they found out few microbiome donors dominated the strain engraftment. These were referred to as “super-donors” and featured a high microbial diversity and a high ratio in *Prevotella* to *Bacteroides* (P/B) strains [160]. Despite these results, Leong and co-workers (2020) found no effects on major metabolic parameters (i.e., insulin sensitivity, liver function, lipid profile, inflammatory markers, blood pressure, body fat percentage, gut health) in obese adolescents treated with a single oral FMT capsule from fecal healthy lean donors, although a reduction in abdominal adiposity was observed [161]. In the near future, clinical trials exploring FMT in adolescents should probably include fiber supplementation that showed to be effective in adult obese patients [47].

An interesting aspect of FMT intervention in obesity would be to explore whether autologous FMT, obtained from the time of maximal loss of weight and transplanted during the phase of weight regaining, might preserve weight loss and glycemic control in moderately obese subjects. To explore this aspect, Rinott and co-workers (2020) enrolled abdominally obese or dyslipidemic participants who received free gym membership together with physical activity and diet guidelines. After six months, during the maximal loss-weight phase, the participants provided a fecal sample that was processed and frozen into capsule form. Afterward, until month 14, a group of the participants ingested autologous oral capsules and another group ingested placebo capsules. Interestingly, only the FMT group that followed the Mediterranean diet and green tea consumption reported attenuation in weight regain (compared to placebo) [162]; the diet induced a modification in the gut microbiota composition during the weight loss phase, and it was able to preserve the weight loss-associated bacterial strains. Moreover, autologous FMT maintained the decreased levels of leptin and cholesterol achieved during the weight-loss phase and it preserved gut microbiota global composition vs placebo [163].

Taken together, the described clinical trials clearly show the beneficial effects of FMT strategies in adults with obesity and metabolic syndromes. Moreover, it opens up new avenues for personalized metabolic attainment preservation in order to maintain weight loss. Strikingly, FMT strategies were exploited in the absence of infections caused by *C. difficile*, again suggesting that FMT can be employed independently from the presence of recurrent infections caused by the pathogen.

### 2.5. Fecal Microbiota Transplantation in Cancer Diseases

FMT-based strategies had been investigated in a few clinical trials in people with cancer conditions. *Melanoma* is a form of skin cancer involving the melanocyte cells that produce the pigment melanin. Recently, new tools exploited the use of immune checkpoint inhibitors to boost a patient’s immune response against the tumor (this therapeutic approach is called “cancer immunotherapy”), and it was found that the gut microbiome interestingly regulates this response [164]. For example, half of the patients with advanced melanoma achieved long-term benefits when treated with monoclonal antibodies targeting the checkpoint controller programmed cell death protein 1 (PD-1) [165,166,167]. Among the variables that could contribute to the success of anti-PD-1 therapy, it has been shown that the gut microbiota modulates anti-PD-1 response. Indeed, as shown in human studies, the gut microbiota of anti-PD-1 therapy responders differs from the non-responders with a higher alpha diversity and an abundance of bacteria of the *Ruminococcaceae* family [168], a higher content in bacterial species *Bifidobacterium longum*, *Collinsella aerofaciens*, and *Enterococcus faecium* [169] and *Akkermansia muciniphila* [170]. Interestingly, if stool samples derived from melanoma patients who positively responded to anti-PD-1 therapy are transplanted into germ-free mice then an amelioration of the antitumor effects of PD-1 blockade occurs, whereas if FMT is performed using stools derived from non-responders then anti-PD-1 failed in its achievement [170]. The latter pre-clinical data obtained in mice were recently replicated in two clinical trials where a subset of patients with metastatic melanoma responded to anti-PD-1 immunotherapy when co-treated also with FMT [171,172]. Thereby, a particular composition of bacterial species in the gut microbiota accounts for the response to anti-PD-1 therapy in people with melanoma and can be transferred by means of FMT.

Treatment of *acute myeloid leukaemia* (AML; a rare but often fatal blood cancer) involves intensive antibiotic and chemotherapies leading to gut microbiota dysbiosis and consequent complications. Malard and colleagues (2021) investigated whether autologous FMT (the use of one’s feces collected during a healthy condition for later in life use to restore gut microflora) corrects the dysbiosis-induced therapies complications and the eradication of drug-resistant bacterial strains. First, the authors confirmed that in their patients’ sample, chemotherapy-induced a decrease in gut microbiota alpha diversity and an increase in proinflammatory strains. After autologous FMT, alpha diversity returned to initial mean levels and microflora composition showed similarity indicating a restoration in its composition [173]. Whether autologous FMT might ameliorate AML condition still has to be investigated, but the clinical trial showed that the treatment is safe and reconstructs microbiota richness and diversity.

FMT is nowadays tested in clinical trials for its efficacy in *urological tumors* resistant to immune checkpoint inhibitors [174]. Moreover, FMT has been proposed also to resolve GI complications after therapy, for example, the common chemotherapy-induced diarrhea in people with metastatic renal cell carcinoma [175] or *C. difficile* infection after bone marrow transplantation in a patient with acute lymphoblastic leukemia [176]. Despite the authors not directly addressing the clinical efficacy of FMT on the carcinoma itself, it still broadened FMT umbrella interventions for cancer diseases.

### 2.6. Fecal Microbiota Transplantation in Auto-Immune, Inflammatory and Infectious Diseases

Rheumatic diseases (or “arthritis”) are autoimmune and inflammatory diseases causing major damage to systems and organs. Among them, *psoriatic arthritis* (PA) presents with inflammation of the joints and enthuses, it is associated with a generally limited quality of life, fatigue, and increased mortality from cardiovascular disease [177]. Despite therapeutic developments, its treatment still remains limited with at least 40% of patients having only a partial response or failing to respond [177,178]. Interestingly, PA is associated with gut microbiota dysbiosis with decreased levels of *Coprococcus*, *Akkermansia,* and *Ruminococcus* strains in comparison to healthy controls [179,180], thus suggesting that restoring microflora diversity might represent a new therapeutic opportunity. Unfortunately, very few clinical trials have been conducted so far, with one by Kragsnaes and co-workers (2021) reporting no major improvements in PA symptoms after FMT [181], despite the patients reporting the treatment to be safe and tolerable and it induced positive changes in their daily life together with renewed hopes for the future [182]. Another research group commented on the failure of the clinical trial claiming that the reason why FMT failed was possibly due to an FMT-induced change in gut microbiota composition that caused a paradoxical triggering of a reactive type arthritis (ReA) disease [183], as suggested by studies in rodent models that link the composition of the gut microbiota with the initiation/progression of the immune-mediated disease [184]. The negative results of the trial could also be dependent on indirect mechanisms of methotrexate that participants received throughout the trial and could not be stopped because of the disease severity. Kragsnaes and co-workers later responded by means of a commentary [185] to McGonagle and colleagues without actually excluding that the paradoxical effect might have taken place during the trial and further claiming that specific clinical trials should investigate the safety, and efficacy of FMT but also unravel similarities and differences in the microbiota dysbiosis and FMT effect mechanisms among patients with different types of inflammatory arthritis. For these reasons, they started a new randomised trial (NCT04924270) where they will investigate in treatment-naïve patients with newly diagnosed conditions (rheumatoid arthritis, ReA, ankylosing spondylitis, PA, gouty arthritis, psoriasis, hidradenitis suppurativa, pulmonary sarcoidosis, Crohn’s disease, and ulcerative colitis) the effect of oral FMT capsules. The estimated study completion date is expected in late 2024 and it should give a clear, detailed big picture of FMT avenues in rheumatic diseases.

*Systemic sclerosis* is an auto-immune rheumatic multi-organ disease characterized by an excessive production of collagen that accumulates in the skin and internal organs and causes damage to the vascular system [186]. Importantly, up to 90% of people with systemic sclerosis experience GI symptoms including diarrhea and gastroesophageal reflux [187], probably caused by an altered gut microbiota composition displaying a lower abundance of anti-inflammatory bacterial genera [188]. Given the limited alternative methods for symptom relief, Fretheim and co-workers (2020) performed the first FMT with commercially available anaerobic cultivated human intestinal microbiota (ACHIM; developed by ACHIM Biotherapeutics AB), namely, a standardized single donor bacterial mixture, in women with cutaneous systemic sclerosis displaying GI symptoms. The patients received the treatment or placebo during gastroduodenoscopy at weeks 0, 2 and 16. The treatment was safe and well tolerated. An amelioration of bloating, diarrhea and fecal incontinence was achieved in most of the patients at week 4 and up to week 16. No changes were observed in SCFAa levels before and after the treatment. As far as fecal microbiota composition is concerned, at week 16 (but not at week 4) beta diversity increased as well as the relative abundance within the *Firmicutes* phylum in three bacterial families *Ruminococcaceae*, *Lachnospiraceae* and *Eggerthellaceae* that are dominant in ACHIM [189]. FMT triggered adaptive immunity measured with a change in the relative abundance of IgA and IgM coated fecal bacteria, thereby showing an interaction between the gut mictobiota and the immune system. Despite the cohort number being limited, the study provided the first clinical efficacy for FMT in systemic sclerosis.

As far as diabetes is concerned, *type 1 diabetes* (T1D) is an auto-immune disease involving autoimmune destruction of the beta cells producing insulin in the pancreas and consequently leading to an excessive level of blood sugar. T-cell targeted strategies aimed to slow down disease progression but only with temporary impact [190]. Given the altered gut microbiota in T1D [191,192], de Groot and colleagues (2021) undertook a randomized controlled trial to assess FMT efficacy on disease progression in recent-onset (<6 months) T1D patients. Three FMT (either autologous or allogenic) were administered via nasoduodenal tube using freshly produced feces at months 0, 2, and 4, and results were collected at months 0, 2, 6, 9, and 12. Overall, FMT reduced the decline in insulin production thereby preserving beta cell function; this was associated with changes in microbiota-derived plasma metabolites and bacterial strains [193]. Another interesting study focused on the role of the gut microbiota in mediating the beneficial effects of exercise on glucose homeostasis. Indeed, exercise can improve insulin resistance but not in all subjects; the phenomenon is known as “exercise resistance”. The mediator seems to be in fact the gut microbiota because only FMT using stools derived from responders to exercise (and not from non-responders) mimics the effect of exercise on insulin resistance in obese mice [194].

Among diseases affecting the skin, *atopic dermatitis* (eczema) is a chronic pruritic condition common in children (up to 20%) less in adults (10%) making the skin red and itchy, it is long-lasting (chronic), it occurs periodically and it importantly impacts patient’s quality of life [195]. Multiple factors are involved, including (i) a dysregulation of the immune system, which brought about the development of the only targeted approved treatment based on the use of monoclonal antibodies [196,197], and (ii) a dysregulation of the skin microbiota. The latter implies a decrease in the diversity of the microbiome that correlates both with the severity of the disease and with an increase in the colonization of pathogenic bacteria [198]. Particular attention was focused on *Staphylococcus aureus*, whose growth suppression by means of topical application on the skin of diluted bleach plus antibiotic [199] or commensal microorganisms that produce potent anti-*S. aureus* molecules, for example, *Staphylococcus hominis*, *Staphylococcus epidermidis,* or *Roseomonas mucosa* [200,201] decreased severity score, pruritus, and corticosteroid use. Early in life gastroenteritis (that affects the gut microbiome) has been proposed as a risk factor for the later development of allergic diseases, including atopic dermatitis [202] calling for a role also for the gut microbiota. Indeed, the gut microbiota can influence the skin via modulation of the immune system, the so-called gut-skin axis [203]. Reddel and co-workers (2019) analyzed the fecal samples derived from children with atopic dermatitis and compared them with those from healthy individuals, and found a dysbiosis characterized by an increase of *Sutterella, Faecalibacterium*, *Oscillospira*, *Bacteroides*, and *Parabacteroides* and a reduction of SCFA-producing bacteria [204], opening up the way for FMT-based interventions. After a recent study in a mouse model showed restoration of gut microbiota upon FMT from control mice (increase in alpha diversity) and immunologic balance (decrease in T-cell activation and white cells counts), increase in SCFAs levels, and decrease in allergic responses [205], Mashiah and colleagues (2021) assessed the clinical safety and efficacy of FMT in adults with moderate-to-severe atopic dermatitis refractory to current treatments. The patients received four FMT treatments in capsule form (one every other week) derived from a single donor and were evaluated one and eight months after the last FMT. Compared to baseline pre-FMT, the authors reported a decrease in corticosteroid usage, a decrease in disease severity, and the patient’s gut microbiota profile becoming similar to the donor’s. Interestingly, the level of similarity correlated with the decreases in disease severity, namely, the more the profile was similar to the donor and the more the severity decreased [206]. Despite the variability of the results obtained in a small cohort of patients, this is the first clinical evidence demonstrating promising FMT-based therapy perspectives for the treatment of atopic dermatitis.

*Multiple sclerosis* is an inflammatory disease of the central nervous system leading to demyelination in the brain and spinal cord by means of an autoimmune mechanism implying a major role for CD4^+^ T helper cells. Among risk factors contributing to the development of the disease, dietary habits are considered potential ones, and the gut microbiota was proposed to be the link between nutrition and inflammatory response because its metabolites can exert a proinflammatory response, regulate T cells and immune gut response [207], thereby promoting demyelination and multiple sclerosis in animal models [208,209]. Since caloric restriction has anti-inflammatory effects in humans [210] and ameliorates inflammation, demyelination, and axon injury in a mouse model [211], Cignarella and co-workers (2018) investigated whether FMT from mice in caloric restriction ameliorates the course of multiple sclerosis in recipient mice. The authors reported a decrease in disease severity and spinal cord pathology compared to mice that received FMT from mice in an ad libitum diet [212], opening up the translation of the approach in human trials as next in line.

Among the *infectious diseases* targeting the immune system, the *human immunodeficiency virus* (HIV) is responsible for immune dysfunctions leading to a chronic inflammatory state and increased mortality [213]. The intestinal mucosal microbiome had been found to play an active role in HIV disease progression with a dysregulation of the intestinal immune barrier that leads to disruption of the intestinal immunity and to dysbiosis characterized by higher levels of *Proteobacteria* and lower of *Bacteroidia*, T cell activation, and chronic inflammation in people infected with HIV [214,215]. In order to tackle the altered mucosal bacterial communities during HIV-1 infection, one clinical trial used a prebiotic oligosaccharide mixture that showed to improve microbiota composition and activation of natural killer cells [216], while another clinical trial exploited the use of synbiotics (a mixture of pre- and pro-biotics) but failed in detecting any major improvements in CD4+ T-cell count, T-cell activation, inflammation, and α and β microbiota diversities [217]. Successively, the latter research group decided to exploit the approach of FMT in a controlled, double-blind, placebo-controlled trial where people with HIV and under antiretroviral therapy received eight rounds of oral FMT capsules (or placebo) and were followed for 48 weeks [218]. Importantly, since HIV infection is associated with decreased butyrate-producing bacteria belonging to *Lachnospiraceae* and *Ruminococcaceae* families [219], and since it has become crystalline in FMT research that the donor microbiota profile is pivotal in achieving engraftment in the host gut and modify the gut microbiota to reach clinical benefits, the authors selected those donors showing enriched butyrate levels in their stools and a predominance of *Bacteroides*, Lachnospiracease, and *Faecalibacterium*. FMT intervention was safe and well-tolerated among the participants. The results indicated that people with HIV receiving oral FMT capsules, in comparison with the placebo group, showed a long-lasting increase in alpha diversity of gut microbiota and an enrichment of several taxa in particular in different members of the Lachnospiraceae family, including *Anaerostipes* spp., *Blautia* spp., *Dorea* spp., and *Fusicatenibacter* spp. There were no changes regarding the circulating CD4+ and CD8+ T cells or the immune activation, but the authors recorded a decrease in the intestinal fatty acid binding protein (IFABP; a biomarker of intestinal injury which predicts mortality). Two participants reported an improvement in chronic constipation. Thus, the study clearly showed that repeated oral FMT capsules derived from selected donors was safe, well tolerated and induced an incremental change in the gut microbiota of people with HIV encouraging further research in the field [218].

FMT finds applications even in the actual coronavirus disease 2019 (COVID-19) pandemic caused by the *SARS-CoV-2 virus*. Indeed, 17.6% of patients developed GI symptoms and 48.1% of patients tested positive for virus RNA in their stools, including those patients that tested later negative for respiratory swaps [220]. Moreover, the gut microbiota changes significantly with an increase of pathogens and a decrease in anti-inflammatory bacteria and those that downregulate the expression of angiotensin-converting enzyme 2 (ACE2; the receptor of SARS-CoV-2) [221]. Given the gut microbiota-targeted positive effects by means of probiotics on GI and clinical symptoms in COVID-19 patients [222], Wu and colleagues (2021) are currently undertaking a clinical trial to establish whether FMT can treat gut microbiota dysbiosis. In particular, they will assess FMT clinical efficacy on GI symptoms, COVID-19 status, recovery from the disease, the inflammatory response, the intestinal mucosal barrier function, as well the clearance time of SARS-CoV-2 from feces [223].

### 2.7. Fecal Microbiota Transplantation in Cardio-Vascular Diseases

Only limited pieces of evidence are available on the role of microbiota and FMT interventions in cardio-vascular diseases and are mostly obtained in animal models [224]. It has been shown a causal link between the gut microbiota–derived metabolite trimethylamine N-oxide (TMAO) and *atherogenesis* [225], namely, the process that brings to the formation of atherosclerotic plaques leading to coronary artery heart disease. Since vegetarians and vegans produce less TMAO (because of their diet) compared with omnivorous subjects, Smits and co-workers (2018) performed FMT by obtaining stools from lean vegan donors and transferring them to patients with metabolic syndrome, but they failed to detect changes in TMAO production or parameters related to vascular inflammation [226].

Based on data on animal models and patients, Zhang and colleagues (2021) recently proposed a causal role of gut microbiota dysbiosis in the elderly in the pathogenesis of *atrial fibrillation*. The authors showed that FMT from aged rats with atrial fibrillation into young rats resulted in higher levels of LPS and higher susceptibility to developing the disease. The authors also found a higher level of circulating LPS in old patients in comparison to younger patients, thereby concluding that the age-related dysbiosis is responsible for an alteration in the microbiota-intestinal barrier-atrial axis and accounts for the disease [227]. Still in animal models, FMT from control mice in the experimental *autoimmune myocarditis* (EAM) mouse model increased microbial richness including an increase in the *Firmicutes*/*Bacteroidetes* ratio, and it ameliorated myocardial injury thanks to reduced inflammation [228]. Despite few pieces of evidence available in the scientific literature on the possible therapeutic value of FMT strategies in cardio-vascular diseases, these data obtained in animal models and in mankind will surely provide a solid ground to undertake further investigations.

### 2.8. Fecal Microbiota Transplantation in Brain Diseases

Gut microbiota surely plays a role in brain functions and dysfunctions [229]. In order to rebalance the gut microbiota in brain diseases, probiotics and prebiotics reported potential [17], but FMT has the greater advantage of transferring eh entire microflora.

First, studies in animal models clarified that fecal microbiota transfer also transfers brain disease-associated features. For example, FMT from normal mice in a mice model of Parkinson’s disease can reduce pathological features in the *substantia nigra* and alleviate physical impairment [230]. Additionally, FMT from people with major depression in normal mice induced behavioral/physiological features characteristic of depression [231]. Finally, FMT from normal mice to an animal model of Fragile X syndrome that also shows autistic-like behavior results in ameliorating autistic-like behaviors [232]. In a mouse model of Alzheimer’s disease, FMT derived from normal control mice improved cognitive functions [233].

As far as studies in humans are concerned, different clinical trials so far investigated FMT in brain conditions, for example, autism spectrum disorders (ASD), Parkinson’s disease, multiple sclerosis, Alzheimer’s disease, and epilepsy [27,30,31]. FMT improved short-term (two months) constipation and motor symptoms in a person with Parkinson’s disease [234] and motor and non-motor symptoms, including constipation, in a larger group of patients [235]. Moreover, FMT improved GI and neurological symptoms in an adolescent with epilepsy and Crohn’s disease [236].

ASD is characterized by impaired social interactions and communication and shares in common the restricted, repetitive, and stereotyped patterns of behavior. Moreover, children and adults with ASD experience GI symptoms [237] that correlate with ASD severity [238,239] and gut microbiota dysbiosis [240,241,242]. Given the short-term effects of probiotics and their limitation in strains availability [17], FMT represents a valuable strategy [243] Kang and colleagues (2017) enrolled 18 children with ASD and administered an initial high FMT dose followed by lower daily doses for 7–8 weeks. The GI symptoms drastically reduced (−80%) at the end of treatment and up to two months. Importantly, behavioral ASD symptoms improved significantly up to two months. The analysis of the bacterial diversity after FMT revealed an increase in bacteria belonging to the taxa of *Bifidobacterium*, *Prevotella*, and *Desulfovibrio* [244]. The research group followed up with the same 18 children with ASD two years after the end of the treatment and found that most of the improvements in GI symptoms were still there together with the improvement of ASD symptoms. Additionally, the authors could still detect those FMT-driven changes in gut microbiota, including an increase in bacterial diversity and relative abundances of *Bifidobacteria* and *Prevotella* [245]. Later, the authors also showed that FMT resulted in distinct fecal and plasma metabolites changes in the recipients [246]. These data highlighted FMT’s potential to achieve long-term benefits on both gut microbiota and GI/ASD symptoms. Li and co-workers (2021) enrolled 40 children with ASD/GI symptoms and 16 control children without GI symptoms. Children with ASD received FMT for eight weeks (via colonoscopy or frozen capsules). FMT improved ASD and GI symptoms and changed the serum levels of neurotransmitters. The authors reported that *Eubacterium coprostanoligenes* were enriched in children with ASD before FMT and decreased after the intervention, thereby identifying *Eubacterium coprostanoligenes* as a therapeutic target to enhance FMT response [247] in children with ASD [248].

FMT has been also exploited to treat *C. difficile* infection in people with neurological disorders, thereby also giving the possibility to study FMT-based strategies to alleviate neurological symptoms [27,30,31,249,250]. Park and co-workers (2021) reported a study on a patient with Alzheimer’s disease diagnosed with severe *C. difficile* infection refractory to antibiotics who underwent two FMT rounds that efficiently eradicate the pathogen and improved GI symptoms. Interestingly, when assessing cognitive functions, the authors also observed a slight improvement in short-term memory, semantic skills, attention, non-verbal learning mood, and expressive affection [251]. In another study with the same dynamic, the patient showed remission from *C. difficile* infection and improvement in cognitive performance [252]. Two studies in a total of four patients with *multiple sclerosis* diagnosed with *C. difficile* infection exploited FMT to resolve infection and constipation, but also reported a progressive improvement in neurological symptoms, including resolution of leg paraesthesia and regaining the ability to walk without assistance [253,254].

## 3. Divergent Clinical Response to Fecal Microbiota Transplantation and the Super-Donor Phenomenon

In each clinical trial discussed so far it was reported that a certain percentage of patients did not respond or responded less to FMT therapy, thereby suggesting that FMT may in fact be limited and reach only a subset of patients. Moreover, the degree of diversity of the donor’s microbiota profile in comparison to the host is pivotal, as suggested already for FMT targeting *C. difficile* infection where heterologous FMT was more successful than the autologous [53,54]. A specific selection of a donor who shows enrichment in those bacteria lacking in the recipient (*Lachnospiraceae* and *Ruminococcaceae*) might represent a good strategy, as exploited in clinical trials on hepatic encephalopathy [38,137]., Moreover, the level of engraftment and long-lasting effect represents the main variables in the successful FMT and clinical outcomes. For example, in IBS the post-FMT engraftment of strictly anaerobic bacteria after FMT does not improve symptoms [255], while post-FMT engraftment in people with chronic constipation is highly populated by species belonging to the *Firmicutes* and carries genes related to polysaccharide metabolism [256]. Certainly, the way of FMT delivery, as well as the number of FMT treatments (for example, one single FMT via colonoscopy is not clinically effective in ulcerative colitis [78]), might play a role as well, but it has become clear that certain donor’s microbiota profiles are more likely to get a better chance to engraft, colonize and to produce a beneficial effect in the host. This may vary in different human conditions. For example, in people with ulcerative colitis specific bacteria in the donor stools were associated with remission or lack of response after FMT [81]; also in another clinical study on people with ulcerative colitis that closely followed up and profiled the colonization and persistence of transferred microbiota along with the transfer of their functions revealed that the persistence of transferred microbes is very variable [257]. Thereby, certain donors possess a microbiota profile with higher chances to achieve successful FMT outcomes than other donors; these donors are referred to as “super-donors” [258].

## 4. Limitations of Current Microbiota Profiling Based on 16S rRNA Gene Sequencing

The majority of evidence summarized in the review comes from clinical studies exploiting 16S rRNA sequencing to characterize the microbiome. Although highly conserved in prokaryotes, the 16S gene contains nine hyper-variable regions (V1–V9), allowing species identification [259]. The most commonly used methods are based on sequencing V3–V4 for a length of less than 300 bp [260]. Other sequencing technologies, such as Oxford Nanopore and PacBIO, can sequence the complete 16S rRNA gene.

In particular, 16S rDNA-based libraries enable targeted species identification by detecting even low levels of DNA in purified samples. Conversely, shotgun sequencing libraries provide a method for the detection of species containing DNA regions that do not amplify with 16S rRNA primers, or whose differences from the template sequence are too high to allow optimum amplification [261,262]. Furthermore, even if DNA polymerases have a high fidelity of DNA replication, basic errors can still occur during PCR amplification. These embedded errors can cause misclassification of the original species [263].

In contrast, a shotgun DNA sequencing library is a DNA library that has been prepared using all the purified DNA extracted from a sample and subsequently fragmented into shorter DNA chain lengths before preparation for sequencing. Taxonomic classification of DNA sequences generated by shotgun sequencing is more accurate when compared to 16S rRNA amplicon sequencing [264]. However, in the shotgun method, there is currently a problem of the high cost to make it routine in the laboratory—those who need to characterize fecal microbiota generally lean towards 16S [265]. In the present review, the discussed studies mostly rely on the 16s-based method technique which constitutes the vast majority of the currently available databases.

## 5. Conclusions and Future Perspectives

In the present review, we presented and discussed a number of clinical studies exploiting FMT to ameliorate both gastrointestinal and disease-related symptoms in a variety of human pathologies ranging from gastrointestinal to liver diseases, from auto-immune to brain diseases. It is clear that (i) gut microbiota dysbiosis populates human pathologies and that (ii) FMT is beneficial to ameliorate disease-related clinical symptoms. Despite the latter points are sound, the precise mechanisms by means FMT induces these beneficial effects are far to be fully elucidated. The general picture is that the gut microbiota does communicate with distant organs via a variety of axes: the gut-skin axis [203], the gut-liver axis [266], and the gut-brain axis [16,17]. Thus, gut microbiota dysbiosis would elicit its effects on the host mainly via modulation of the endocrine and the immune systems. Conversely, reinstatement of gut microflora using FMT would reinstate the physiological modulation of these systems.

A number of clinical trials described in the present review were performed independently on the presence of infection by *C. difficile* and when patients were refractory to the current therapeutic strategies, suggesting that since FMT is safe and tolerable could be used even as the first choice to ameliorate the clinical symptoms in a number of human pathologies. More and more clinical trials are emerging testing the safety and efficacy of FMT even in other diseases, for example, that in primary sclerosing cholangitis, a liver disease with no effective medical therapies but associated with gut microbiota dysbiosis, and in which the authors decided to explore FMT just for the simple fact that “FMT has been reported to restore the microbiome in other disease states” [129]. Despite the challenging task of recruiting suitable fecal donors for FMT [267], thanks to the accepted safety (only mild adverse events during or shortly after treatment were reported so far), we may imagine that in the future more and more clinical trials will exploit FMT strategies to ameliorate clinical symptoms and conditions on a number of other human conditions independently on *C. difficile* infections. In the near future, the accurate identification and subsequent characterization of super-donor gut microbiomes will pay the way for the establishment of stool microbiota banks for the treatment of a variety of human conditions.

## Figures and Tables

**Figure 1 jcm-11-04119-f001:**
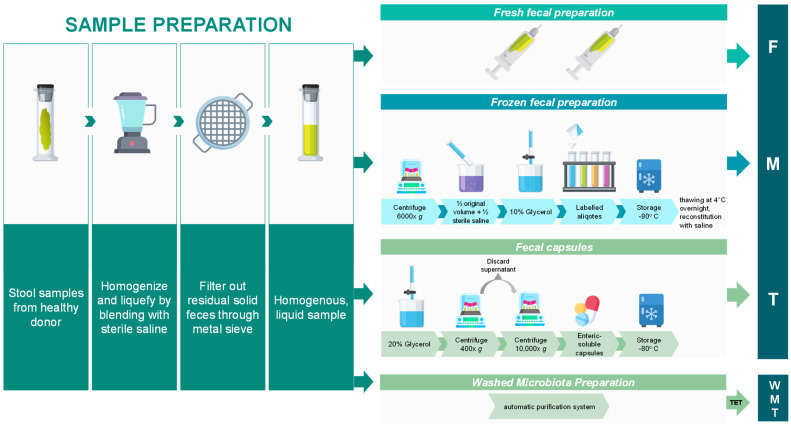
Stools sample preparation for fecal microbiota transplantation. Fecal stools (100–150 g) dissolved in saline solution (NaCl, 0.9%) are homogenized and larger particles removed by filtration. The fresh fecal sample can be used within less than 6 h for FMT. Alternatively, the fresh fecal preparation is further processed with multiple steps of filtration, cry-protected in glycerol (10%), frozen, and kept at −80 °C for later use. The preparation of FMT capsules involves the addition of freeze-drying protectant glycerol (20%), centrifugation (400× *g*), the supernatant is discarded and it is centrifuged again at high speed (10,000× *g*); the sediment is incorporated into an enteric-soluble capsule and stored at −80 °C [35]. Alternatively, the material can be lyophilized (vacuum dried) to obtain fecal powder inserted in capsules, and stored at −80 °C for later use. If multiple steps of microfiltration, centrifugation, and suspension are carried out using an automatic system, it is referred to as washed microbiota preparation (WMP). WMP transfer via colonic transendoscopic enteral tubing (TET) is referred to as washed microbiota transplantation (WMT).

**Figure 2 jcm-11-04119-f002:**
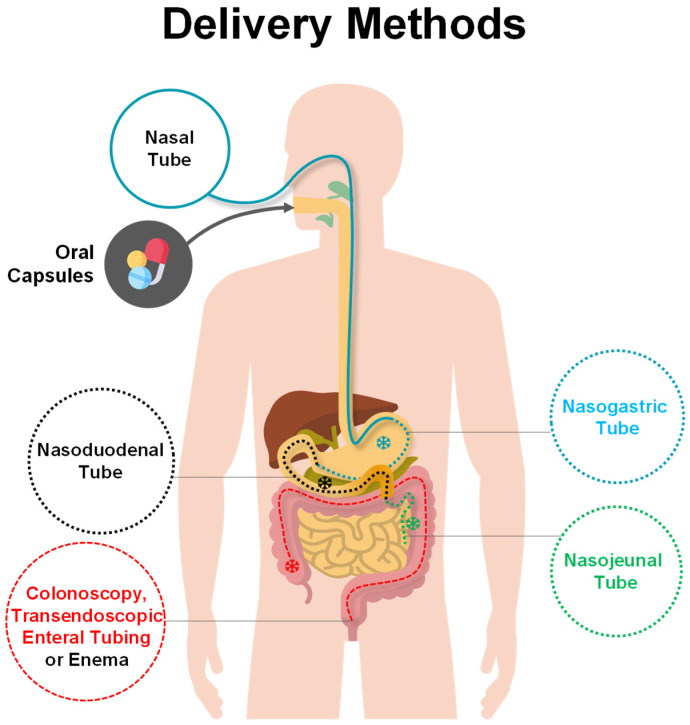
Delivery methods for fecal microbiota transplantation. The delivery methods are classified into upper gastrointestinal routes, including nasogastric/nasoduodenal/nasojejunal tubes and capsules, and lower gastrointestinal routes, including enema colonoscopy and colonic transendoscopic enteral tubing.

**Figure 3 jcm-11-04119-f003:**
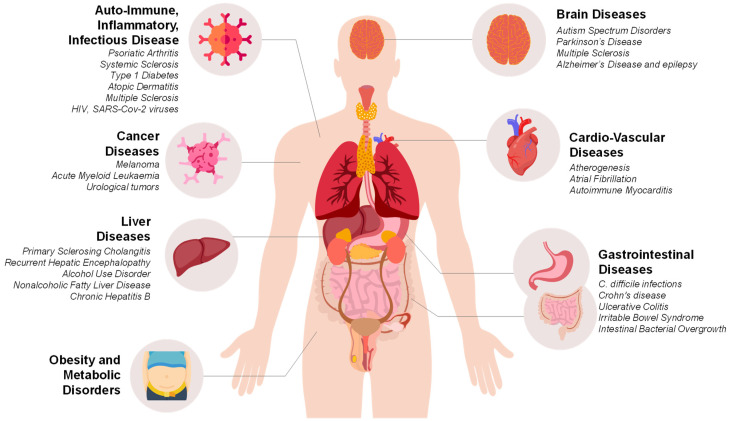
Fecal microbiota transplantation in human diseases. Schematic cartoon summarizing fecal microbiota transplantation-based clinical trials in the different human diseases. Refer to the text for details.

## Data Availability

Not applicable.

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
