# Peer review of "Fecal Microbiota Transplantation as New Therapeutic Avenue for Human Diseases"

_jcm, 2022, doi:10.3390/jcm11144119_

Round 1
Reviewer 1 Report
The authors provided a good review making significant contribution to understanding the development of FMT as a therapeutic strategy.
One minor suggestion is one section as the conclusion of mechanism/pathways involved and correlating safety issues could be added in the end of this manuscript.
Author Response
RESPONSE TO REVIEWER 1
We wish to thank Reviewer for the time, effort and attention dedicated in considering and reviewing our manuscript. In fact, reviewer’s comments and suggestions were very helpful and we do feel that the quality of our manuscript improved after revision.
Together with the point to point response to the reviewer, we are submitting a new version of the manuscript addressing ALL reviewers’ comments (in track change). Please note that in the revised manuscript the main changes and amendments requested by the reviewer had been made in trach change mode and highlighted in red in order to facilitate reviewer’s revision. English corrections have been done only in track change mode.
Reviewer’s comments are in italic.
Q1: The authors provided a good review making significant contribution to understanding the development of FMT as a therapeutic strategy.
We are pleased that the reviewer feels that we provided “a good review making significant contribution to understanding the development of FMT”.
Q2: One minor suggestion is one section as the conclusion of mechanism/pathways involved and correlating safety issues could be added in the end of this manuscript.
We thank the reviewer for her/his insightful comment. We addressed this point adding a new paragraph on page 25 of the revised manuscript in the “Conclusion and future perspectives”, now Section 5.

Reviewer 2 Report
In their paper “Fecal Microbiota Transplantation as New Therapeutic Avenue for Human Diseases” Biazzo and Deidda comprehensively review the current knowledge on FMT, from sample preparation and delivery methods to pieces of evidence in support of FMT in different diseases.
I read the review with great interest. While the different pieces of evidence are detailed, I found the way the information is proposed to be dispersive. This is particularly true in paragraph 2.2. This part is extremely long and highlights aspects such as the safeness and the no-lasting effect of FMT already proposed earlier in the review. In my opinion, shortening this paragraph by retaining only the critical aspects will improve the delivery and make the text more engaging.
One aspect that, in my opinion, should be considered with more care is the description of the microbiota that should not be limited to the Bacteria Superkingdom (from line 46). The authors should also comment on the limitations of profiling the microbiota by performing only 16S rRNA gene sequencing.
(Lines 59-74) The crosstalk between the microbiota and the immune system should be acknowledged.
(Line 86) Bacteriophages’ description was already introduced in lines 55-56.
Revision of the English is recommendend mainly to shorten the numerous too long sentences. Small mistakes widespread in the manuscript should also be fixed, e.g. in lines 98-100, 112, 223, 247, 285, 317, 341, 345, 402, etc.
From line 866, it should be SARS-CoV-2 and not SARS-cov-2.
Author Response
RESPONSE TO REVIEWER 2
We wish to thank Reviewer for the time, effort and attention dedicated in considering and reviewing our manuscript. In fact, reviewer’s comments and suggestions were very helpful and we do feel that the quality of our manuscript improved after revision.
Together with the point-to-point response to the reviewer, we are submitting a new version of the manuscript addressing ALL reviewers’ comments (in track change). Please note that in the revised manuscript the main changes and amendments requested by the reviewer had been made in trach change mode and highlighted in red in order to facilitate reviewer’s revision. English corrections have been done only in track change mode.
Reviewer’s comments are in italic.
In their paper “Fecal Microbiota Transplantation as New Therapeutic Avenue for Human Diseases” Biazzo and Deidda comprehensively review the current knowledge on FMT, from sample preparation and delivery methods to pieces of evidence in support of FMT in different diseases.
We are pleased that the reviewer feels that we “comprehensively reviewed the current knowledge on FMT”.
I read the review with great interest. While the different pieces of evidence are detailed, I found the way the information is proposed to be dispersive. This is particularly true in paragraph 2.2. This part is extremely long and highlights aspects such as the safeness and the no-lasting effect of FMT already proposed earlier in the review. In my opinion, shortening this paragraph by retaining only the critical aspects will improve the delivery and make the text more engaging.
We thank the Reviewer for his/her comment. When we designed the structure of the Review, we were concerned by the complexity of the topic that could become dispersive. Thereby, in order to provide more clarity in the organization of the information given in the review, we decided to divide the different sections based on the physiological system under investigation (gastrointestinal system, nervous system and so on) and we did provide Fig. 3 to help through the reading.
The point is that most of the clinical trials exploiting FMT were focused so far on the gastrointestinal system. This obviously impacted on the length of the paragraph 2.2. We now shortened to an extent the paragraph based on reviewer’s suggestion
One aspect that, in my opinion, should be considered with more care is the description of the microbiota that should not be limited to the Bacteria Superkingdom (from line 46). The authors should also comment on the limitations of profiling the microbiota by performing only 16S rRNA gene sequencing.
This is a very good point and we thank the reviewer for his/her insightful suggestion. Based on reviewer’s suggestion we now inserted a full new section (Section 4) addressing these points on page 24 on the revised manuscript.
(Lines 59-74) The crosstalk between the microbiota and the immune system should be acknowledged.
We acknowledged the crosstalk between the microbiota and the immune system on page 2 on the revised manuscript.
(Line 86) Bacteriophages’ description was already introduced in lines 55-56.
We removed these lines.
Revision of the English is recommendend mainly to shorten the numerous too long sentences. Small mistakes widespread in the manuscript should also be fixed, e.g. in lines 98-100, 112, 223, 247, 285, 317, 341, 345, 402, etc.
We perfomed a wide revision of the English in order to shorten long sentences. We fixed the small mistakes mentioned by the reviewers across the manuscript. These changes are made in track-changes mode..
From line 866, it should be SARS-CoV-2 and not SARS-cov-2.
We fixed this.
